# Neural-Attentional Architectures for Deep Multi-Agent Reinforcement Learning in Varying Environments

**Matthew A. Wright** [1]   **Roberto Horowitz** [1]

## Abstract

Many potential applications of reinforcement learning (RL) in the real world involve interacting with other agents whose numbers vary over time. We propose new neural architectures for these multi-agent RL problems. In contrast to other methods of training an individual, discrete policy for each agent and then enforcing cooperation through some additional inter-policy mechanism, we propose learning multi-agent relationships at the policy level by using an attentional architecture. In our method, all agents share the same policy, but independently apply it in their own context to aggregate the other agents' state information when selecting their next action. The structure of our architectures allow them to be applied on environments with varying numbers of agents. We demonstrate our architecture on a benchmark multi-agent autonomous vehicle coordination problem, obtaining superior results to a full-knowledge, fully-centralized reference solution, and significantly outperforming it when scaling to large numbers of agents.

## 1. Introduction

Multi-agent reinforcement learning (RL) is said to be much more difficult than conventional, single-agent, RL. In addition to the typical obstacles in single-agent RL (like temporal credit assignment due to sparse rewards and navigating the exploration-exploitation tradeoff), multi-agent RL adds complications such as an intrinsically higher dimensionality, per-agent credit assignment, and (from the perspective of each individual agent) environmental nonstationarity during the learning process (i.e., if multiple interacting agents are all learning at the same time, then one agent's knowledge about how others react to their actions quickly becomes

outdated) (Hernandez-Leal et al., 2018).

Many real-world problems where the application of RL has been proposed can be classified as multi-agent problems. Autonomous vehicle planning and control, for example, has been considered as a natural domain for RL due to the difficulty of forming a complete first-principles model of the driving environment for the application of classic control (Shalev-Shwartz et al., 2016; Wu et al., 2017b). Many of the complications in autonomous driving come from needing to reason about and interact with other agents; e.g., other vehicles and pedestrians. Even assuming human-level perception, these multi-agent interactions are the source of inefficiencies. For example, it is well-known that inter-vehicle dynamics, as drivers react to other drivers' actions, and those reactions are reacted to, etc., often lead to traffic congestion and suboptimal outcomes for all involved (Sugiyama et al., 2008). Advocates of autonomous vehicle technologies argue that they will improve transportation safety and efficiency by harmonizing these inter-vehicle behaviors. In fact, several authors have recently demonstrated this potential by showing how a small number of computer-controlled vehicles can dissipate stop-and-go congestion waves (Cui et al., 2017; Stern et al., 2018; Vinitsky et al., 2018; Wu et al., 2017b). Effective solutions to multiagent RL can help autonomous vehicles deliver on their promised gains.

An additional complexity that arises in real-world multi-agent scenarios like driving is that, from an "ego" agent's perspective, the number of other agents varies. Of the aforementioned works, three (Cui et al., 2017; Stern et al., 2018; Wu et al., 2017b) only consider artificial environments where the number of other vehicles is fixed. The fourth (Vinitsky et al., 2018) applies RL to more realistic multi-agent coordinative vehicle control problems where the number of agents varies, but unsatisfyingly relies on a central coordinator, and observes that performance degrades when the RL policy is asked to coordinate the actions of a large number of vehicles. In real-world applications, it would be desirable to obtain a control policy that is effective for varying numbers of agents, and can improve performance when the effective action space (e.g., when there are more vehicles that can coordinate to mitigate congestion)

---

[1] University of California, Berkeley, California, USA. Correspondence to: Matthew A. Wright <mwright@berkeley.edu>.

$36^{th}$ *International Conference on Machine Learning*, Workshop on RL for Real Life. Long Beach, California. Copyright 2019 by the author(s).

expands.

In this paper, we present a new framework for deep multi-agent RL that attempts to address those problems. Our proposed method centers on the application of new neural network architectures specific for the multi-agent setting. In particular, we apply neural attention (Bahdanau et al., 2015; Vaswani et al., 2017) as a fundamental building block in our learning model. We argue that this framework has appealing properties: among other benefits, it provides a principled solution to the per-agent credit assignment problem and can be flexibly applied to situations with varying numbers of agents.

The remainder of this paper is organized as follows. In section 2, we briefly review the mathematical framework of RL in general and multi-agent RL. Section 3 discusses a particular benchmark problem in coordinated autonomous vehicle control (Vinitsky et al., 2018) that we use as a framing problem. After a discussion of some of the needs for applying RL to complex multi-agent problems like autonomous vehicle coordination and our method's promise towards meeting those needs in section 4, section 5 reviews neural attention and discusses its application on the problem introduced in section 3. Section 6 provides exhaustive implementation details for our application. Section 7 presents our preliminary results and discusses how they and our implementation compare to the reference RL implementation from (Vinitsky et al., 2018). Finally, section 8 summarizes next steps in developing our attention-based approach to deep multi-agent RL.

## 2. Multi-Agent Reinforcement Learning: Background

### 2.1. The general RL setting

RL is typically presented in the mathematical framework of finite-time, discounted Markov decision processes (MDPs) (Duan et al., 2016). These MDPs are defined by a tuple $(S, A, P, r, \rho_0, \gamma, T)$, where $S$ is the state space, $A$ is the action space, $P : S \times A \times S \to \mathbb{R}_{\geq 0}$ is the transition probability distribution, $r : S \times A \to \mathbb{R}$ is the reward function, $\rho_0 : S \to \mathbb{R}_{\geq 0}$ is the probability distribution on initial states, $\gamma \in (0, 1]$ is a reward discounting factor, and $T$ is the time horizon. The goal is to maximize the cumulative discounted reward $\sum_{t=0}^{T} \gamma^t r(s_t, a_t)$ where $s_t$ and $a_t$ are the state and action, respectively, at time $t$.

In the RL problem, the probability distributions and/or the reward function are unknown. The objective is to learn a *policy* $\pi : S \times A \to \mathbb{R}$ that maximizes the expectation of the discounted future reward, $E \sum_{t=0}^{T} \gamma^t r(s_t, a_t)$. Traditionally, the policy is assumed to be stochastic, i.e., a probability distribution, and is written $\pi_\theta(a_t|s_t)$ where $\theta$ is a parameter vector that parameterizes the policy. The objective is then

to find the optimal parameter vector $\theta^*$, defined as

$$\theta^* = \arg\max_\theta E_\tau \sum_{t=0}^{T} \gamma^t r(s_t, a_t) \qquad (1)$$

where $\tau = (s_0, a_0, s_1, a_1, \dots)$ is a shorthand for the entire trajectory, $a_t \sim \pi_\theta(a_t|s_t)$, and $s_{t+1} \sim P(s_{t+1}|s_t, a_t)$.

Most difficulties in RL stem from the fact that $P$ and $r$ are unknown, but the solution to (1) is fundamentally dependent on both of them. Methods to solve RL problems iterate on their solution candidate $\theta$ for many iterations. Typically, it is desired that, in early stages of the solution process, $\pi_\theta$'s are chosen that can be used to gather information about the form of $P$ and $r$; and the $\pi_\theta$'s obtained in the end stages of the solution process leverage the information gained to solve (1) using the learned approximations of $P$ and $r$. These two sub-goals are often referred to as the "exploration vs exploitation" problem.

The entity that draws actions $a_t$ from $\pi_\theta(a_t|s_t)$, executes them, and observes the resulting sample $s_{t+1} \sim P(s_{t+1}|s_t, a_t)$ and reward value $r(s_t|a_t)$ (for the particular $s_t, a_t$) is typically called the "agent."

In modern deep RL, the policy $\pi_\theta$ is expressed by a deep neural network, with $\theta$ being the neural network weights. Development of particular algorithms to iterate on the neural network weights $\theta$ in deep RL has been a topic of much research in recent years (Duan et al., 2016; Henderson et al., 2017).

### 2.2. Multi-agent RL

So far, we have just described the background to traditional, non-multi-agent RL. Multi-agent RL, as its name suggests, adds complications by having multiple agents. Let $\mathcal{I}$ denote the set of agents. Typically these agents are considered to have discrete policies $\pi_\theta^i$, where the superscript indexes individual agents $i \in \mathcal{I}$. The multi-agent RL problem is to obtain the optimal policy $\pi_{\theta*}^i$ for either some or all of the $i \in \mathcal{I}$ (e.g., all of the agents when the agents are cooperating, but some subset of $\mathcal{I}$ when the agents are competing, e.g., learning to play opposing sides of a competitive game).

The review paper (Hernandez-Leal et al., 2018) notes that the natural approach of training each agent independently, iterating on each agent's $\pi_\theta^i$ using samples $(s_{t+1}^i, s_t^i, a_t^i, r_t^i)$ as outlined in section 2.1, is likely to fail in the multi-agent case. This is because the unknown transition distribution $P$ is now a function of *every agent*'s state and action, i.e.,

$$P(s_{t+1}|s_t, a_t) = \prod_{i \in \mathcal{I}} P(s_{t+1}^i|s_t, a_t) \qquad (2)$$

where $s_t$ and $a_t$ in (2) now denote the set of states and actions for all agents $i \in \mathcal{I}$: $s_t = \{s_t^i : i \in \mathcal{I}\}$, $a_t =$

$\{a_t^i : i \in \mathcal{I}\}$. That is, every agent is interacting with the environment and each other at the same time.

The great difficulty in the multi-agent learning process comes from the fact that, typically, every agent is learning at the same time. This means that, from the point of view of a single agent $i$ that only has control over its own action $a_t^i$, the optimal $\theta^{*i}$ from (1) is now a function of the other agents' actions $a_t^j, j \in \mathcal{I} \setminus \{i\}$. Those actions are of course dependent on the other agents' policies $\pi_\theta^j, j \in \mathcal{I} \setminus \{i\}$. If all agents are updating their policies $\pi_\theta^i$, then $P$ (2) changes continuously.

The reward function in the multi-agent case will differ based on whether the agents are cooperating or competing. If the agents are purely cooperating, then the reward $r(s_t, a_t)$ will be functions of the same set $s_t, a_t$'s from (2). If the agents are competing, then different rewards are given to each agent or team of agents.

Defining *local* rewards for each agent, $r(s_t^i, a_t^i)$, can make the learning process easier by decoupling then, but in many RL problems defining them is infeasible (Hernandez-Leal et al., 2018). A major thrust of multi-agent RL research is thus to construct methods to relate each agent's $(s_i^t, a_i^t)$ to the global reward $r(s_t, a_t)$. This is called the per-agent credit assignment problem and is a central problem in multi-agent RL.

### 2.3. Approaches to multi-agent RL

Many authors have proposed approaches more sophisticated training approaches to overcome the difficulties just mentioned. The review paper (Hernandez-Leal et al., 2018) notes three trends: 1) encouraging agents to learn how to communicate information to other agents, 2) encouraging agents to learn behaviors that are inherently cooperative, and 3) encouraging agents to form an internal model of other agents' policies. In general, these approaches retain the idea of training individual policies per agent, but adjust the training goal to include context-specific multi-agent information. This adjustment is done, by, for example, altering the reward function, adding auxillary learning tasks or auxillary NN modules to solve the aforementioned problems specific to multi-agent settings. These auxillary modules tie the per-agent NNs together, and are to propagate information about other agents back to each individual one. (Rashid et al., 2018) note that these supervisory coordinating modules describe a problem setting where the agents can be trained concurrently in a controlled environment where the global information for the auxillary modules is available. At test time, the agents will be deployed without these auxillary modules, and will hopefully exhibit multi-agent-aware behaviors obtained during the controlled training. For more details on specific techniques, see (Hernandez-Leal et al., 2018).

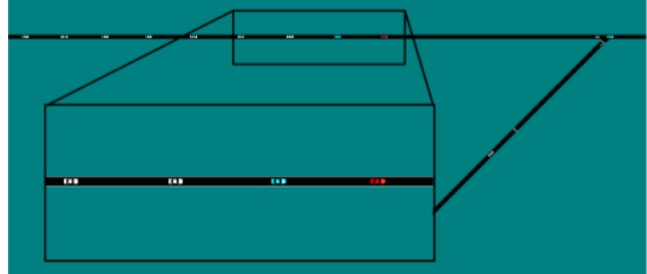

*Figure 1.* "Merge" benchmark road network, with zoom-in to show simulated vehicles. From (Vinitsky et al., 2018).

### 2.4. Our approach

We propose a new approach to multi-agent RL: *learning how to relate individual agents' policies both to a) the global reward, and to b) each other (at the policy level), via neural attentional architectures in both value and policy networks.* Unlike the prior works that introduce new coordinative modules or cooperation-inducing rewards, our framework changes the architecture of the agent policies $\pi_\theta$. To give the explanation of our method a more concrete base, we next discuss a particular motivating multi-agent RL problem.

## 3. Our Framing Problem

In this work, we frame our discussion against a benchmark multi-agent RL problem introduced by (Vinitsky et al., 2018) (shown in Figure 1). The work proposed several multi-agent reinforcement learning problems based on mixed-autonomy traffic (road traffic with mixtures of autonomous and human-driven vehicles). We will consider the "Merge" problem. In this problem, two single-lane roads merge into one. At the merge, the vehicles will compete for space, inducing congestion and a high social cost. The RL problem is to take control of some subset of the vehicles and dissipate this congestion.

For each controlled vehicle, the state space $\mathcal{S}$ is a five-dimensional vector of its own speed and the speed and bumper-to-bumper gap of the immediately preceding and following vehicles. The action space $\mathcal{A}$ is the controlled vehicles acceleration (a scalar value). The problems reward function encourages all vehicles to move quickly, while having the controlled vehicles maintain not-too-small inter-car distances.

In (Vinitsky et al., 2018), the canonical solution uses a single-agent approach rather than a multi-agent approach. There, a central controller receives all observations, stacks them into one vector, and computes all actions. However, the number of controlled vehicles on the network will change as they enter and exit, so to use a traditional single-agent

MLP (successive fully-connected neural network layers) architecture, a fixed number of vehicles to control (five, in this case), and the network-wide observation vector is either truncated or zero-padded as needed. On the action end, if there are fewer than five controllable vehicles present, extra actions are discarded, and when there are more than five, some are left uncontrolled.

## 4. Why Our Method?

The original proposers of the "merge" benchmark (Vinitsky et al., 2018) note that the reference solution has several shortcomings. Most critical is the "unfixed" state and action spaces. In most RL problems, including multi-agent RL problems, the state and action spaces are of fixed size. The state space may be, for example, readings from a fixed number of sensors, an image with a constant number of pixels, etc., and the action dimension a fixed number of actuators. Even in complex multi-agent RL problems like computer strategy games (Hernandez-Leal et al., 2018), there exist a fixed number of agents, making the joint state and action spaces fixed in dimension.

In contrast, in the "merge" problem (and, indeed, in many coordinative transportation control tasks like using CACC to form platoons dynamically), the number of agents varies over time. In the merge problem, this happens as controllable vehicles enter and exit the network. The benchmark's proposers (Vinitsky et al., 2018) note that the reference solution of controlling at most five vehicles effectively throws away extra information when more than five vehicles are present. Although unmentioned, the padding and truncation likely also makes the learning problem harder because the RL agent is expected to learn by itself to not assign credit to the ignored actions (without knowledge that they have been ignored), making the credit assignment problem even more difficult. One solution is to train different policies for different numbers of agents and select between them as the situation changes, but training many policies would, among other issues, vastly increase the RL sample requirements. In contrast, our proposed method seeks to be cross-trainable by allowing valid backpropagation for any number of agents.

(Vinitsky et al., 2018) also note that the fully-centralized controller is "unlikely to be possible in real road networks." A method to decentralize RL training and execution is a critical step towards its deployment to real transportation networks. Our method has an advantage in this area in that, since it is valid for any number of agents, it can by construction be executed by a single agent in a fully decentralized manner. This means that, to the best of our knowledge, this paper represents the first work on a decentralized multi-agent solution to the mixed-autonomy traffic problems presented in (Vinitsky et al., 2018).

As of this writing, our method is immature, and (as we will discuss near the end of the following section) our neural architecture is lacking many features of state-of-the-art attentional methods, so results that deliver to RL the same performance boost as attention did to machine translation (Bahdanau et al., 2015; Vaswani et al., 2017) are not presented here. Nevertheless, given the appealing properties outlined in this section, it is worth investigating how the attention framework can be adapted from supervised learning to RL at all. The next section covers our efforts towards this.

## 5. Attentional Architectures for RL

### 5.1. Overview

Neural attention (Bahdanau et al., 2015; Luong et al., 2015; Vaswani et al., 2017) is a powerful neural architectural technique. It allows a deep neural network to learn how to relate distinct entities (e.g., in an RL context, distinct agents) and their associated data to each other, and, to assign importance to these relations. For example, several high-profile papers ((Bahdanau et al., 2015; Vaswani et al., 2017), etc.) have discussed how, in a language translation problem, an attention module in a neural network appeared to assign importance to word pairings that appear important to meaning (e.g., assigning high importance from the position of a pronoun, to the noun to the pronoun is referring).

Many overviews of the intricacies of neural attention exist (e.g., (Bahdanau et al., 2015; Battaglia et al., 2018; Luong et al., 2015; Vaswani et al., 2017), and others). We give only a brief summary here, and refer to the particulars of the "merge" problem discussed in section 3.

Suppose that at time $t$, there exist $|\mathcal{I}(t)|$ controllable vehicles. Rather than stacking the $|\mathcal{I}(t)|$ per-vehicle states $s_i(t) \in \mathbb{R}^5$, $i \in \mathcal{I}(t)$ into a $|\mathcal{I}(t)| \cdot 5$-dimensional vector, then then padding or truncating it to a fixed size, we leave the observations as an $|\mathcal{I}(t)| \times 5$ tensor, with the first dimension being dynamic. This sort of variable-size tensor is a valid input into (self-) attention-type layers, unlike a traditional fully-connected layer.

The output of our attention-structured deep neural network is a dynamically-sized tensor whose shape is $|\mathcal{I}(t)| \times \eta$, with $\eta$ being the parameters that parameterize a single vehicle's stochastic policy $\pi$, e.g., if a vehicle's stochastic policy is a Gaussian parameterized by a mean and variance that are themselves functions of the state and parameterized by the neural network weights (symbolically, $a_t^i \sim \mathcal{N}(\mu_\theta(s_t), \sigma_\theta(s_t))$), then $\eta = 2$.

Next, we dive into some more details on the particulars.

## 5.2. Architecture Details

We use an attention-structured deep network that will always return $|\mathcal{I}(t)|$ actions, while allowing each vehicle's action to depend on others' state information. At its most fundamental level, the structure of an attentional neural network layer uses two subnetworks: one for *embedding* and one for *aggregation*. Let $i \in \mathcal{I}(t)$, and say that $s_i \in \mathbb{R}^5$ is the state associated with vehicle $i$. Then, define $f : \mathbb{R}^5 \to \mathbb{R}^m$ as the embedding subnetwork, and $g : \mathbb{R}^5 \times \mathbb{R}^5 \to \mathbb{R}$ as the aggregation subnetwork. Then $h_i \in \mathbb{R}^m$, (we use $h$ here to denote this layer output vector in light of the fact that it is usually a "hidden layer" in a deep neural network) the output of an attention layer corresponding to that vehicle, is

$$h_i = \sum_{j \in \mathcal{I}(t)} \text{softmax}(g)_j f(s_j) \tag{3}$$

where by $\text{softmax}(g)_j$ we mean the $j$th entry of the softmax of a logit vector whose $j$th logit is $g(s_i, s_j)$. The idea is that the embedding subnetwork $f(s_i)$ learns to map each vehicle's state into a useful representation, and the aggregation subnetwork learns, as a function of both $s_i$ and $s_j$, the relative usefulness of the $f(s_j)$'s for the encoding in $h_i$.

In this way, the output of the attention layer has a discrete element for each agent, but each element contains information from all agents.

In the results presented here, we use the "scaled dot-product" attention layer of (Vaswani et al., 2017), where $f(\cdot)$ and $g(\cdot, \cdot)$ are made of matrix multiplications. We also make use of the relative position embeddings of (Shaw et al., 2018) and multi-headed attention (Vaswani et al., 2017). Multi-head attention allows the attentional layer to learn multiple independent embeddings of the input data. (Shaw et al., 2018)'s relative position embeddings add a learned bias vector to both the $f(s_j)$ and $g(s_i, s_j)$ computations, where the bias vector is different for different $(i, j)$ relationships. In this work, we use (Shaw et al., 2018)'s relative position embeddings such that the bias vector used depends on the relative position upstream or downstream of $j$ to $i$. There is one bias vector for $i = j$, one for $i$ being the next-most-downstream controlled vehicle from $j$, etc. We clip the representations at a distance of three vehicles; all $i, j$ pairs of a relative distance of three or more share the same bias vector.

In this work, we use 4 attention heads of 16 units each. The output of the attention layer is thus a tensor of dimension $b \times |\mathcal{I}(t)| \times m$, where $b$ is the batch dimension, $\mathcal{I}(t)$ is the number of vehicles, and $m = 64$ (4 heads times 16). This tensor is passed through a fully-connected hidden layer with 64 units (each of the $b \cdot |\mathcal{I}(t)|$ attention layer outputs pass through this layer identically and in parallel). Both the attentional and fully-connected sublayers are followed by a ReLu nonlinearity and a layer-normalization operation (Ba et al., 2016) (with learned scale and location parameters), in that order.

The output of the above layers then goes into the output layer, whose output parameterizes the stochastic policy. In this work, our stochastic policy is a per-agent Gaussian distribution with mean and log-variance computed by the same fully-connected layer for each agent. The same layers are used for all vehicles $i \in \mathcal{I}(t)$, and can be computed fully in parallel.

This structure of attentional sublayer followed by shared-over-agents fully-connected sublayer is inspired by the Transformer architecture of (Vaswani et al., 2017), though we use only one such layer and omit any residual connections.

## 5.3. Attentional Proximal Policy Optimization

Proximal Policy Optimization (PPO) methods (Schulman et al., 2017) are a popular class of RL training algorithms. One attractive quality of PPO methods is their relative simplicity compared to other RL training algorithms. For this reason, in this work we use PPO for evaluating our attentional architectures.

The traditional PPO algorithm assumes only a single agent. In this section, we describe some necessary generalizations to be able to apply PPO to our attentional multi-agent architectures.

One general PPO objective function of a policy parameter vector $\theta$ at timestep $t$ is of the form (Schulman et al., 2017)

$$L_t^{PPO}(\theta) = E\left[L_t^{CLIP}(\theta) - c_1 L_t^{VF}(\theta) + c_2 S\left[\pi_\theta\right](s_t)\right]$$

where

$$L_t^{CLIP}(\theta) = E\Big[\min\Big(r_t(\theta) \cdot \hat{A}_t,$$
$$\text{clip}(r_t(\theta), 1 - \epsilon, 1 + \epsilon) \cdot \hat{A}_t\Big)\Big]$$

with $r_t(\theta) = \frac{\pi_\theta(a_t|s_t)}{\pi_{\theta_{old}}(a_t|s_t)}$ the ratio of the likelihood of the actually-taken action $a_t$ under $\theta$ to the likelihood under $\theta_{old}$, the initial value of $\theta$, $\text{clip}(\cdot, \cdot, \cdot)$ a clipping function that clips $r_t(\theta)$ within $\epsilon$ of 1, $\hat{A}_t$ an estimate of the advantage at time $t$, and where $L_t^{VF}$ is the squared error of an estimate of the value function $V(s_t)$, $S[\pi_\theta](s_t)$ is the entropy of the policy distribution outputted by the neural network for input state $s_t$, and $c_1$ and $c_2$ are constants.

The key insight that allows us to apply PPO to the attentional multi-agent architecture is that every term is a function of the *policy* $\pi_\theta$ (and the action(s) taken by that policy) rather than any particular *agent*. More specifically, given a set of per-agent states and rewards $(s_t^i, a_t^i), i \in \mathcal{I}(t)$ and a scalar reward $r(t)$, we can define the *policy* distribution $\pi_\theta(s_t)$ as

simply

$$\pi_\theta(s_t) = \prod_{i \in \mathcal{I}(t)} \pi_\theta^i(s_t) \qquad (4)$$

and the *policy* likelihood of $a_t | s_t$ as just

$$\pi_\theta(a_t | s_t) = \prod_{i \in \mathcal{I}(t)} \pi_\theta^i(a_t^i | s_t). \qquad (5)$$

where by $\pi_\theta^i(s_t)$ we mean the distribution generated by the parameter vector $\theta$ when viewing $s_t$ from agent $i$'s perspective. Using (4) and (5), one can compute all terms involving the policy distribution in the PPO objective, as well as an estimate of the Kullbeck-Leiber divergence between between two $pi_\theta$'s in a straightforward matter.

Architecturally, generating an estimator of the scalar value function $V_\theta(s_t)$ from per-agent deep embeddings of each agent's perspective of $s_t$ is less principled. In this work, we use a value network with identical architecture to the policy network described above, add an agent-wise max pooling operation at the end, and pass the output of that through a fully-connected layer to produce a scalar value estimate.

### 5.4. Attention's Real-World Applicability

It is worth noting a few details that make the attentional architecture appealing for multi-agent RL problems. Of key importance is that each agent's actions is computed fully in parallel. What this means is that each agent can actually compute its action locally, independent of the other agents, using only its knowledge of its and the other agents' states. While we discussed the above computations in terms of tensors batched over agents, in practice this batching is only for purposes of computational parallelism and ease of explanation.

Also of note is how using the attentional architecture allows for the straightforward application of a simple and relatively well-understood *single-agent* RL training algorithm (namely, PPO). As noted, this is because, technically speaking, the actions and reward are conditioned on the *policy*, which is held fixed across agents (showing this with appropriate mathematical rigor is part of ongoing work). The question of how each agent needs to reason about all other agents when determining its own action is made part of the end-to-end learning problem. The ability to deploy classic RL algorithms like PPO, as opposed to needing multi-agent-specific RL algorithms like QMIX (Rashid et al., 2018) is noteworthy.

Since all the agents use the same policy, we may think about each agent's state and action, and the states of the other agents, as an individual training example for the single policy. It seems that the only obstacle to a fully-decentralized training regime, where gradients can be computed locally, is the fact that to estimate the scalar reward, we need to aggregate encoded information over agents in our value network by, e.g., our max-pooling. However, since all agents share the same policy, we should be able to assume that any agent with knowledge of the others' states can perform an estimate not only of its own action, but also the others'. This means that the value function is fully estimable locally, by each agent. While in our current setting, the presence in the PPO loss of an action log-likelihood term (for us, that means the joint log-likelihood of all agents' actions given their per-agent policy parameters) prevents a clean break to fully-decentralized training, such an effort is an important part of future work.

## 6. Implementation Details

The "merge" baseline described above is implemented in the framework *Flow* (Wu et al., 2017a), which is a Python codebase built on the widely-used microscopic vehicle traffic simulator SUMO (Krajzewicz et al., 2012) that adapts SUMO to the widely-used RL problem standard "env" developed in OpenAI *Gym* (Brockman et al., 2016). We implemented our neural network architecture in *Ray*, (Moritz et al., 2017). In particular, we modified Ray's implementation of Proximal Policy Optimization (PPO) (Schulman et al., 2017) to be compatible with the network architecture we described above.

All PPO hyperparameters (training minibatch size, training epochs, Generalized Advantage Estimator parameter $\lambda$, MDP discount factor $\gamma$, Adam learning rate, PPO entropy coefficient, PPO clipping parameters) were left as the the same as in the reference solution (Vinitsky et al., 2018).

We also used Ray to produce a reference solution similar to (Vinitsky et al., 2018)'s that used a single-agent policy with the padding and truncation discussed in section 3. For our single-agent reference, we use a two-hidden-layer policy with 64 units in each hidden fully-connected layer and a tanh nonlinearity in between, a common policy architecture in the RL literature (Henderson et al., 2017). This 64x64 architecture serves as a comparison to the attentional architecture that has the same number of hidden units. We note that the original authors of the reference solution used a neural network of greater depth.

Our code, taking the form of modules for Ray, is available online at `github.com/mawright/attn_rl`.

## 7. Results

(Vinitsky et al., 2018) proposed several different configurations of the "Merge" problem, varying in the penetration rate of autonomous vehicles and the maximum number of vehicles that are allowed to be controlled. At the low end, "Merge 0" requires the control of at most 5 vehicles, and

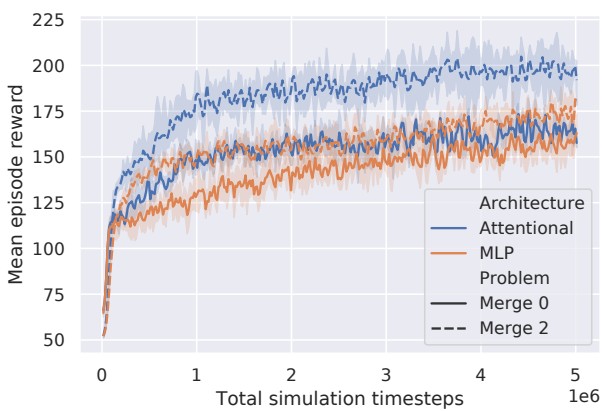

*Figure 2.* Learning curves for PPO on the "Merge 0" and "Merge 2" benchmarks of (Vinitsky et al., 2018). "Merge 0" requires the control of up to 5 vehicles, and "Merge 2" the control of up to 17. The mean and 95% confidence interval of episode reward over four runs for each architecture are shown.

on the high end, "Merge 2" requires the control of up to 17 vehicles.

Figure 2 shows learning curves for PPO on the "Merge 0" and "Merge 2" benchmarks, for both our attentional architecture and the reference MLP architecture. On both problems, we obtain superior performance to the reference architecture. While our gain in performance at first glance appears modest, it must be emphasized that the single-agent MLP implementation is a high bar to clear. The single-agent MLP acts as a global coordinating controller. In contrast, during execution our attentional architecture truly acts in a distributed manner, with each agent's action computed independently. The ability of our distributed controller to outperform a centralized controller is likely due to the extra difficulty in per-agent credit assignment introduced in the single-agent problem by padding and truncation, as discussed in section 4. This perhaps also explains why our attentional policy improves significantly on the Merge 2 problem relative to the Merge 0 problem, while the MLP policy's performance gain when enjoying more than three times as many degrees of freedom is more modest (for the MLP, the enhancement of its action space comes with a far greater degree of potential padding).

## 8. Conclusion

We proposed attentional architectures for deep multi-agent RL. Our architectures present principled solutions to several important problems in multi-agent RL. First, using attention allows for the use of a single policy for multiple numbers of agents, by making each agent's local aggregation of the other agents' states part of the end-to-end learning problem.

Second, attention allows the application of classic "single-agent" RL training algorithms like PPO, obviating the need for multi-agent-specific training regimes. Third, the sharing of policies among agents allows for a greater degree of decentralization in RL training and execution. Future work should explore the extension of both of these points to move towards greater contextual transferability and decentralized coordination in deep RL.

## Acknowledgements

This research was supported by the National Science Foundation under grant CPS-1545116 and Berkeley DeepDrive. We also made use of the Savio computational cluster provided by the Berkeley Research Computing program at the University of California, Berkeley.

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
