# OpenReview forum: "Neural-Attentional Architectures for Deep Multi-Agent Reinforcement Learning in Varying Environments"
_ICML.cc/2019/Workshop/RL4RealLife — Submitted to RL4RealLife 2019_

### Official Review · AnonReviewer1 · 2019-05-23
**Ambitious idea but no supporting results**

**Rating:** 1
**Confidence:** 5

**Review:**

This paper considers the “merge” problem in the autonomous driving scenario where human and autonomous driving vehicles are mixed in the road. The authors propose to use the attention network in order to solve the credit assignment problem in multi-agent reinforcement learning. Even though the authors have a very promising idea for a well-designed problem, the manuscript does not have sufficient results to support their proposal. As the authors admit it at the end of Section 4, the method that is used in the simulation is very immature and not modified for the current scenario. In a current form, the manuscript is more proposal than a research article. The reviewer recommends putting more efforts to develop their amazing idea and submit again in the near future.

---

### Official Review · AnonReviewer2 · 2019-05-23
**Comments**

**Rating:** 2
**Confidence:** 3

**Review:**

This paper proposes a decentralized multi-agent algorithm. In particular, the paper adds an attentional architecture to PPO and applies it to a mixed-autonomy traffic problem. It is an interesting idea. However, it seems to be an ongoing work and the experiments fail to show me the advantage of the proposed algorithm.

Cons:
- Centralized training of decentralized policy is a standard paradigm in multiagent RL. Thus I would suggest the author add more baselines in the experiment, such as MADDPG and COMA.
- I struggle to see how the model actually benefits from the attention mechanism. It would be great if the author can do some visualization in terms of network structure and the attention during execution.

---

### Decision · Program_Chairs · 2019-05-28

Reject